# Exploring the Relationship between Cover Crop Adoption and Soil Erosion Severity: A Case Study from the Simcoe Watershed, Ontario, Canada

Katherine Shirriff, Krishna Bahadur KC *[ID] and Aaron Berg [ID]

Department of Geography, Environment, and Geomatics, University of Guelph, Guelph, ON N1G 2W1, Canada; kshirrif@uoguelph.ca (K.S.); aberg@uoguelph.ca (A.B.)
* Correspondence: krishnak@uoguelph.ca; Tel.: +1-519-824-4120 (ext. 58189)

**Abstract:** Runoff from agricultural fields during the nongrowing season is a significant factor leading to phosphorous loading and diminishing water quality in Lake Simcoe, Ontario. Cover crops offer the potential to alleviate phosphorous loss during the nongrowing season by minimizing soil erosional processes and uptaking excess phosphorous; however, recent research suggests that its adoption remains relatively low. More concern lies with the lack of cover crop adoption on areas that are sensitive to soil erosion. This study intends to investigate the likelihood of agricultural productions located on erosive soils to adopt cover crops. Using satellite imagery in corroboration with the Universal Soil Loss Equation (USLE), this study reveals the frequency of cover crop production and associates soil loss sensitivity at a 30 m resolution from 2013 to 2018. Consistent with recent literature, this study reveals that a small portion (18%) of agricultural operations in the south Simcoe Watershed have incorporated cover crops over the past six years. Cover crops tend to be adopted at a low frequency in areas that have a low sensitivity to soil erosion. This study reveals that areas with higher soil erosion sensitivity are consistent with low-frequency adoption, indicating that these areas are less likely to adopt cover crops regularly. Promoting farm-scale benefits associated with cover crops should target areas in the south Simcoe Watershed that are prone to soil erosion to mitigate total phosphorus (TP) loading into Lake Simcoe.

**Keywords:** soil erosion severity; cover crops; phosphorus; soy corn rotation





## 1. Introduction

Depleted water quality and eutrophication is a growing concern in Ontario's freshwater lakes, particularly in Lake Simcoe. The degraded freshwater environment in Lake Simcoe is associated with elevated phosphorous loads from surrounding urban and agriculture areas [1]. The Ontario Ministry of the Environment and Climate Change [1] reports that approximately 25% of total phosphorus (TP) entering Lake Simcoe originates from nearby agricultural operations, the majority of which occurs during the nongrowing season (Figure 1) [2,3] as soils are left bare and vulnerable to wind and water erosion. During the nongrowing season, the temperate winter climate, including midwinter and spring thaws, promotes high discharge and, in turn, TP loss on agricultural fields [2]. Lake Simcoe is subject to such trends as peak discharge and TP loads from the surrounding subwatersheds occur in March and April (Figure 1)—indicative of spring thaw [4]. In combination with discharge data from the Government of Canada [5] and TP loading data collected by [6], Figure 1 reveals average discharge and TP loading rates into Lake Simcoe from four subwatersheds during the nongrowing season (October–April).

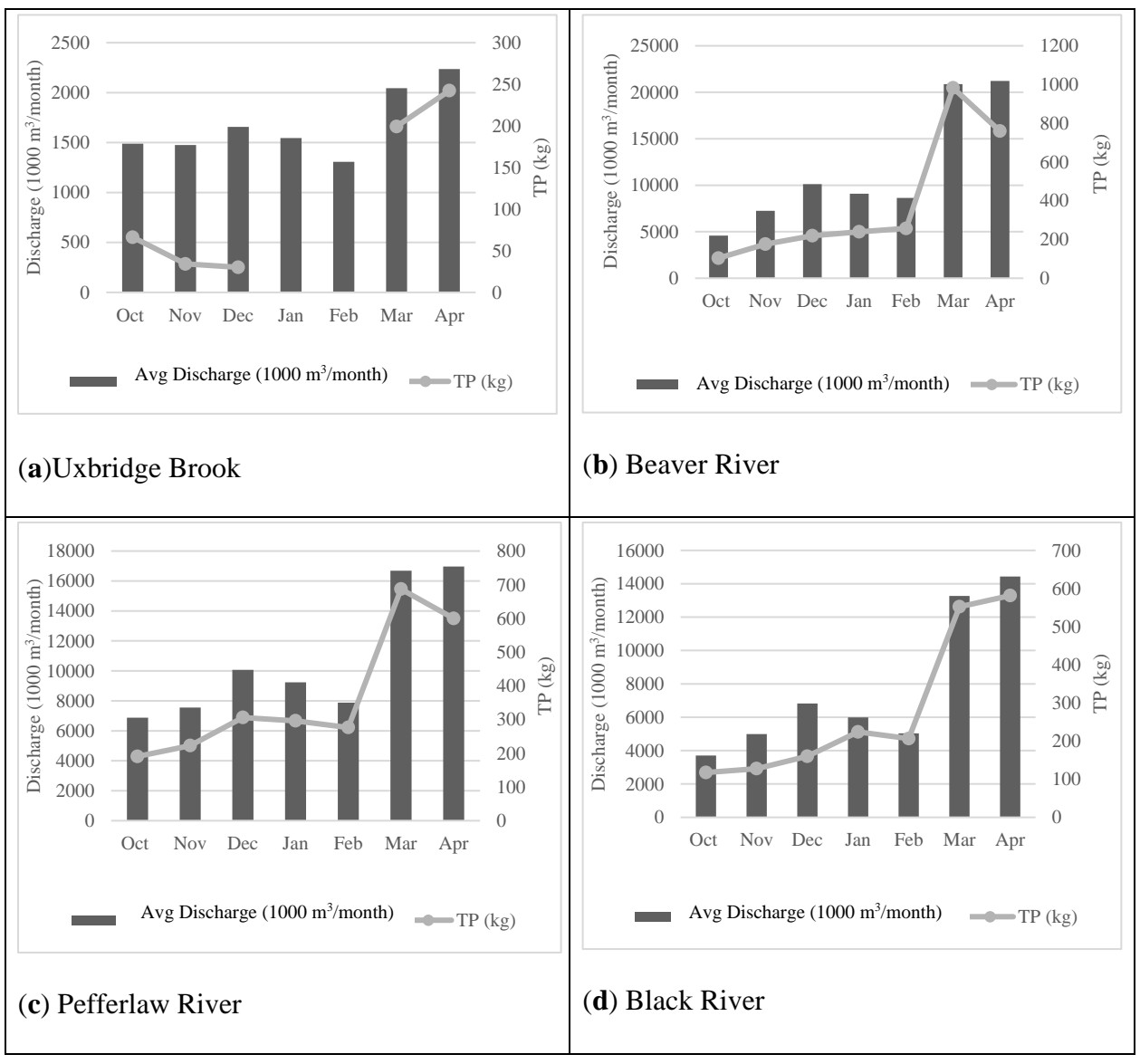

**Figure 1.** Average discharge (1000 m$^3$/month) (Government of Canada, 2020) with total phosphorus (TP) (kg) from 2011–2016 for the Uxbridge Brook (**a**), Beaver River (**b**), Pefferlaw River (**c**), and the Black River (**d**) [6].

The use of cover crops is an attractive management practice to reduce soil erosion and TP loading during the nongrowing season [7–9]. Despite the magnitude of the potential benefit, research indicates a low adoption rate of cover crops across the Corn Belt—including Ontario [10,11]. The lack of adoption of cover crops is of concern, as it suggests that fields are left bare during the nongrowing season. Even more concerning is that low adoption rates can extend into areas that are prone to soil erosion. Cover crop practices should target agricultural operations located on erosion-prone soils within the Simcoe Watershed to control TP loss from runoff during the nongrowing season. Currently, there is a need to identify areas that are prone to soil erosion and assess the likelihood of cover crop practices in the Simcoe Watershed. This study intends to fill this research gap by analyzing cover crop adoption trends in correlation to areas that are prone to soil erosion.

Collecting ground cover data is a popular technique to monitor land use and classification. This method is useful and accurate at field-level scales; however, given spatial and temporal variability of farm practices and land classifications on larger, regional scales, this method requires a lot of time and money to collect such extensive ground cover data.

Satellite imagery is an effective and efficient approach for collecting such data on a large scale. This research utilizes the Annual Crop Inventory (ACI) dataset provided by Agriculture and Agri-Food Canada (AAFC)—a dataset that identifies all land use types at a 30 m resolution. The ACI dataset is derived from optical (Landsat-8, Sentinel-2) and radar (RADARSAT-2) base satellite images [12]. This study used ACI datasets using ESRI ArcGIS to analyze cover crop adoption trends over the south Simcoe Watershed.

The Universal Soil Loss Equation (USLE) is used in this study to investigate soil erosion over the Simcoe Watershed at a 30 m resolution. The USLE and its derivatives—the Revised (RUSLE) and Modified (MUSLE)—are widely accepted models for estimating the sensitivity of soil erosion of certain areas based on the slope gradient, a soil erodability factor, rainfall intensity, cropping management, and support practice factors [13–15]. KC [16] used such a model to produce erosion-susceptible maps for an area that has suffered due to shifting cultivation located in the mountainous regions of Northern Thailand. The KC [16] study utilized remote sensing in corroboration with GIS methods to calculate the USLE and further classify pixels into different soil erosion severity levels. A similar study by Bartsch et al. [17] investigated soil loss and transport at Camp Williams in northern Utah—identifying areas that are sensitive to soil loss in a region notorious for intense summer storms and subsequent sediment loading of riparian zones. Bartsch et al. [17] utilized the RUSLE to create an erosion-risk classification map by classifying the quantitative values generated using the RUSLE and grouping them into classes. To make the USLE and its derivatives more user-friendly to map, a study by Zhang et al. [13]) integrated the MUSLE with ArcGIS to create a tool to identify runoff, peak flow, and soil loss for a rainfall event within a watershed. This model requires compatible layers, including Digital Elevation Model (DEM), Soil Layer, Rainfall Layer, and Land Cover Layer. Zhang et al. [13] ran their application using data for Black Hawk County, Iowa, USA, and generated a useful map for policymakers to identify problematic areas for erosion.

Currently, there is a significant amount of research examining cover crop adoption benefits for TP control throughout the nongrowing season [2,3,7–9]. Further, there is a body of literature identifying adoption trends of cover crops in the Corn Belt [10,11]. However, a better understanding of the likelihood of farmers adopting cover crops based on soil erosion would help to understand if cover crops are well-targeted to erosion-prone areas. To date, the degree of how well-targeted cover crop adoption is toward areas where significant erosion can take place is not known. This study intends to investigate this correlation in the Simcoe Watershed. This paper utilizes AAFC's ACI dataset to identify cover crop adoption in corroboration with the USLE of the south Simcoe Watershed to investigate a correlation between soil erosion and cover crop adoption. The objective of this study is to use USLE to identify hot spots for soil erosion and identify the degree to which cover crops have been adopted in these sensitive regions.

## 2. Methods

### 2.1. Study Site

Lake Simcoe is a large (722 km$^2$ surface area) freshwater lake consisting of the central basin, Cooks Bay, and Kempenfelt Bay located on the south and west sides of the lake, respectively. All analyses in this study were conducted within the Simcoe Watershed, focusing on field crops grown within agricultural fields in the study site (Figure 2). The study analyzed five subwatersheds, located south of Lake Simcoe; these include the Uxbridge Brooke, Beaver River, Maskinonge River, Pefferlaw River, and Black River subwatersheds, all of which discharge into Lake Simcoe, except for the Maskinonge River that discharges into Cooks Bay. Soils across these five subwatersheds are prime for agriculture, as the landscape comprises of Class 1, 2, and organic soils [18]. The soil types include silt loam, clay loam, loam, sand, and loamy sand. Agriculture occupies approximately 48% of the area across the five subwatersheds. Fallow, corn, and soybean occupy most of the agricultural production in the study site, representing 23.6, 10.9, and 10.2% of the agricultural area in the study site, respectively.

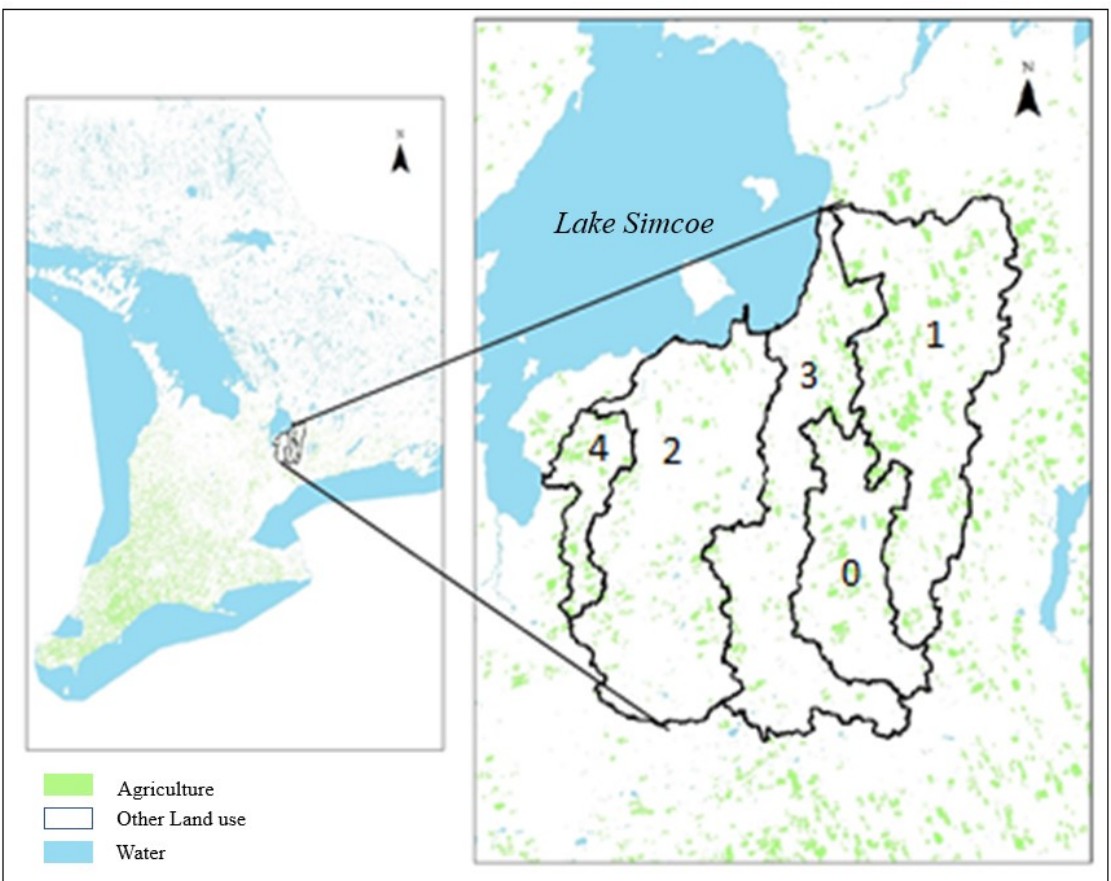

**Figure 2.** Agricultural areas shown in green across the Uxbridge Brooke (0), Beaver River (1), Maskinonge River (2), Pefferlaw River (3), and Black River (4) subwatersheds. Image manipulated from AAFC annual crop inventory [12].

Across the five watersheds, the average winter and summer temperatures are −4.7 °C and 18.1 °C, respectively. The coldest temperature occurs in January with an average of −7 °C and the hottest temperature occurs in July with an average of 19.9 °C. Annual rainfall and precipitation across this study site average 728.7 mm and 886.3 mm, respectively.

## 2.2. Universal Soil Loss Equation (USLE)

Soil erodibility sensitivity was calculated in this study using the Universal Soil Loss Equation (USLE)—a popular soil erosion model created by Wischmeier and Smith (1978 NBY NUMBER 14). This model explores five factors that influence soil erosion:

$$A = R * K * LS * C * P \qquad (1)$$

Variable A represents the average amount of soil loss in an area—measured in tonnes $ha^{-1}year^{-1}$, R represents the rainfall erosivity factor, K represents the soil erodability factor, LS is an index for slope length and steepness factors, C represents cover and management, and P represents support practice (Equation (1)). The USLE was calculated and mapped across the study site at a 30 m resolution. The generated map classifies the calculations into five soil loss rates (tonnes/ha/year) thresholds, based on OMAFRA [19] USLE classifications: very low (0–6.7), low (6.7–11.2), moderate (11.2–22.4), high (22.4–33.6), and severe (>33.6).

### 2.2.1. Rainfall Erosivity Factor (R)

The Rainfall Erosivity Factor (R) represents the erosive potential of rain due to the intensity of rainfall events. This factor requires continuous site-specific rainfall intensity

data measured in 30 min increments. Such data is difficult to acquire over a large study, as a plethora of weather stations are required to collect continuous and detailed rainfall data. This barrier was experienced in this research, as it encapsulates an extensive study site. Instead, to fulfill the R factor requirement for the USLE, OMAFRA [19] estimates an R-value of 90 for the Toronto Station and Tweed Station data. OMAFRA's [19] R-value estimation encapsulates the regions within this study site. This value remained constant across the five subwatersheds in this study.

### 2.2.2. Soil Erodibility Factor (K)

The soil erodability factor (K) represents the vulnerability of specific soil types to erosional forces based on the soil texture and soil organic matter (OM) content. Soil types were mapped on ArcGIS using the Soil Landscapes of Canada Version 3.2 [20,21] for the study site. Based on the sand, silt, clay, and percent of OM content of each soil type, a K factor was assigned to each cell using the OMAFRA [19] K factor data.

### 2.2.3. Slope Gradient Factor (LS)

The slope of any given area is an essential factor for understanding the risk of erosion, as steeper slopes influence erosional forces. In this study, the soil loss equation utilizes the slope length (L) and steepness (S), providing the slope gradient factor (LS). The Sediment Transport Index (STI) was used to calculate the slope gradient factor, as it characterizes the erosion and depositional processes [22] for any given watershed based on the topography. The index is used under the assumption that the contributing area (As) is directly related to discharge and the slope [23]. The Sediment Transport Index is defined as:

$$STI = (m + 1) * (A_s/22.13)^m * (\sin \beta / 0.0896)^n \tag{2}$$

The Provincial Digital Elevation Model (PDEM) dataset provided by OMAFRA [24] was derived to solve for the slope and flow accumulation across the study site at a 30 m resolution using Esri ArcMap. The generated slope and flow accumulation represent the $\beta$ and $A_s$ variables, respectively. The contributing area exponent $m$ and the slope exponent $n$ were assigned the constant values of 0.4 and 1.3, respectively, as they are proven to be the best fit for the soil loss equation [25]. The STI at a 30 m resolution was generated for the study site using White Box Geospatial Analysis Toolbox [23] across the study site.

### 2.2.4. Cover Management Factor (C)

Reducing erosion of a given area is dependent on land cover and management. The Cover Management Factor (C) determines the effectiveness of soil and crop management systems for reducing erosion and is defined as the ratio between soil loss under specific crops with the equivalent soil loss in continuous fallow and tilled land [15]. Generally, the C factor ranges between 0 and 1, indicating very strong vegetation cover and barren land, respectively. Much recent research has estimated the C factor using remote sensing data through the Normalized Difference Vegetation Index (*NDVI*), as it positively correlates to the C factor [26–31]. *NDVI* values were calculated and averaged.

$$NDVI = \frac{NIR - IR}{NIR + IR} \tag{3}$$

where *NIR* is the spectral reflectance in the near-infrared band and *IR* is the spectral reflectance in the red band. The *NIR* and *IR* bands were extracted from Landsat 8 images of the study site. *NDVI* values range between −1 and +1, where values closer to +1 represent green vegetation, 0 represents bare soil, and values less than 0 represent bodies of water.

*NDVI* values were calculated at a 30 m resolution for six Landsat 8 images obtained monthly from April—September 2018. The six *NDVI* images were averaged and the resulting *NDVI* values were scaled to approximate C-values using the following formula:

$$C = \exp\left[-\alpha * \left(\frac{NDVI}{\beta - NDVI}\right)\right] \tag{4}$$

where $\alpha$ and $\beta$ are parameters that determine the shape of the *NDVI-C* curve. Values 2 and 1 are assigned to parameters $\alpha$ and $\beta$, at the recommendation of Van der Kniff et al. [30]. Figure 3 reveals the relationship between *NDVI* and C.

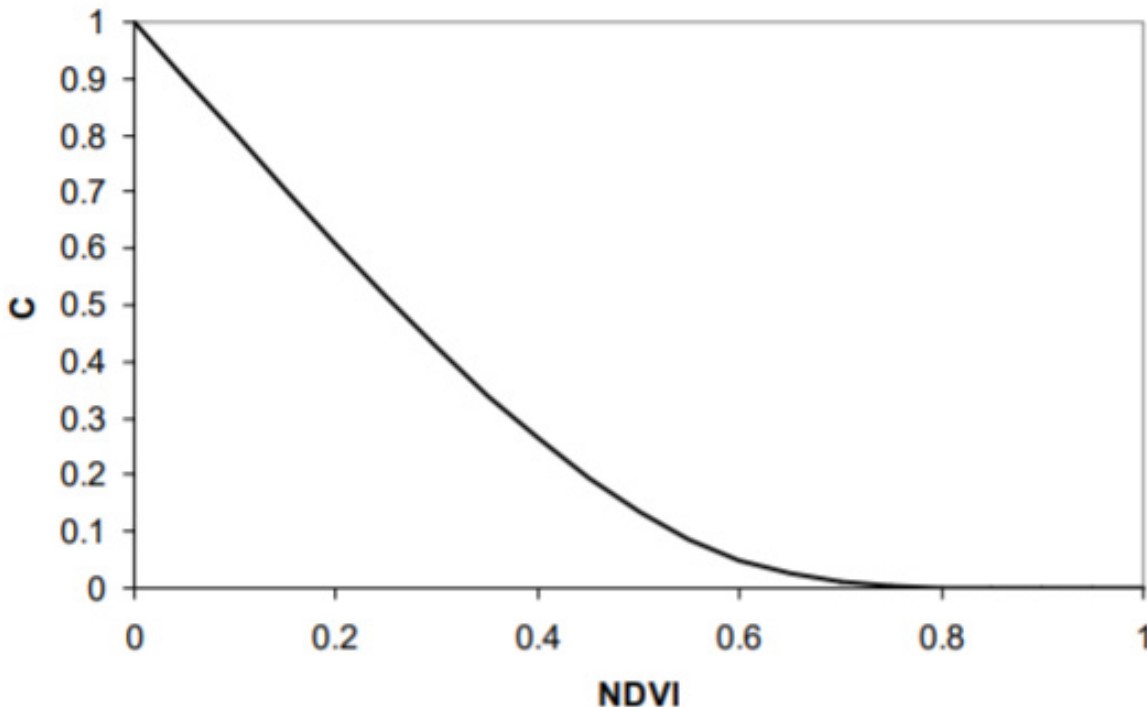

**Figure 3.** Relationship between Normalized Difference Vegetation Index (*NDVI*) and Cover Management Factor (C) using exponential scaling formula [31].

### 2.2.5. Support Practice Factor (P)

The support practice factor reflects the effects of practices that will reduce the amount of erosion, including strip contouring, straight row farming up and down the slope, cross slope cultivation, and strip cropping [19]. The *p*-value ranges between 0 and 1, where 0 represents very good erosion control and 1 represents no erosion control solutions. Consistent with [32–34], the chosen value of *p* = 1 was selected, as obtaining conservation practices data for estimating P was not consistently available for the areal extent of this study site.

### 2.2.6. Application: Esri ArcMap

The USLE model was generated using ESRI ArcMAP, using five factors (L, S, K, R, C) as input data. These variables were identified using elevation, soil properties, rainfall, and vegetation data. All datasets were stored at WGS_1984_UTM_Zone_17N projection. Figure 4 reveals the Esri ArcMAP workflow, modeling the USLE.

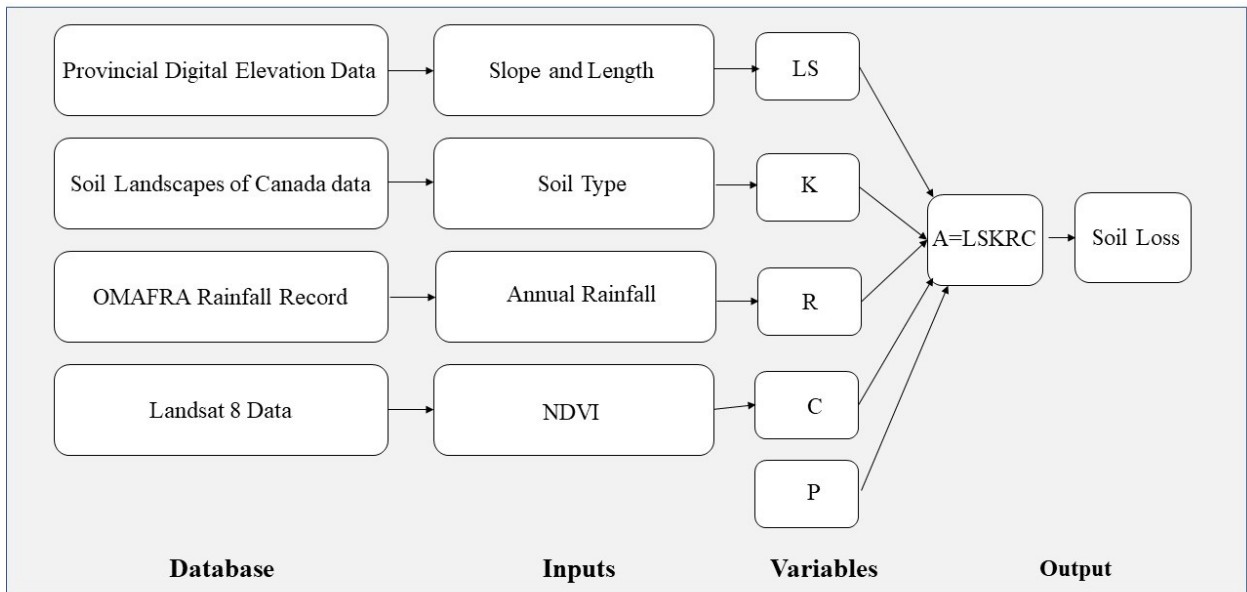

**Figure 4.** Flow chart of Universal Soil Loss Equation (USLE) model using ArcMap.

*2.3. Cover Crop Frequency Analysis*

AAFC's ACI dataset identifies and classifies all land cover types, with a focused differentiation on agricultural production, at a 30 m resolution. This study utilizes ACI data to investigate cover crop adoption from 2013–2018 in the Simcoe Watershed. ESRI ArcMap was used to define the study site boundary (Figure 1) and reclassify ACI datasets from 2013–2018. OMAFRA [35] identifies rye, oat, and winter wheat as some common cover crops grown in southern Ontario. This study acknowledges that rye, oats, and winter wheat are not always grown as cover crops; however, for the purposes of this study, they will be identified as cover crops. Additionally, this study does not investigate double-cropping systems or off-season management cover crops, such as red clover, due to remote sensing limitations. Using Esri ArcMAP, rye, oats, and winter wheat classifications were collapsed into a cover crop category, and all other land uses were reclassified as null data.

The six reclassified ACI layers were added together to generate a new layer, revealing the frequency of cover crop appearance for each pixel over the six years. The identified pixels containing cover crops were superimposed onto the generated USLE map and assigned the associated USLE value. Stacked bar graphs were generated to reveal trends between cover crop frequency and soil erosion sensitivity.

## 3. Results

*3.1. USLE Analysis*

Based on the integration of LS, K, R, C, and P factors of the USLE, five classes of annual soil loss sensitivity indexes were identified. Previous government research in Ontario (OMAFRA, 2012) identifies these thresholds as: very low (0–6.7 tonnes/ha/year), low (6.7–11.2 tonnes/ha/year), moderate (11.2–22.4 tonnes/ha/year), high (22.4–33.6 tonnes/ha/year), and severe (>33.6 tonnes/ha/year) (Figure 5). The areal extent of the USLE reveals that the majority of the study site equates to very low soil loss class, representing 86.6% of the area; low, moderate, high, and severe represent 6.4, 5.0, 1.2, and 0.8%, respectively (Figure 5 and Table 1). Table 1 reveals the areal extent of soil loss sensitivities at the subwatershed scale.

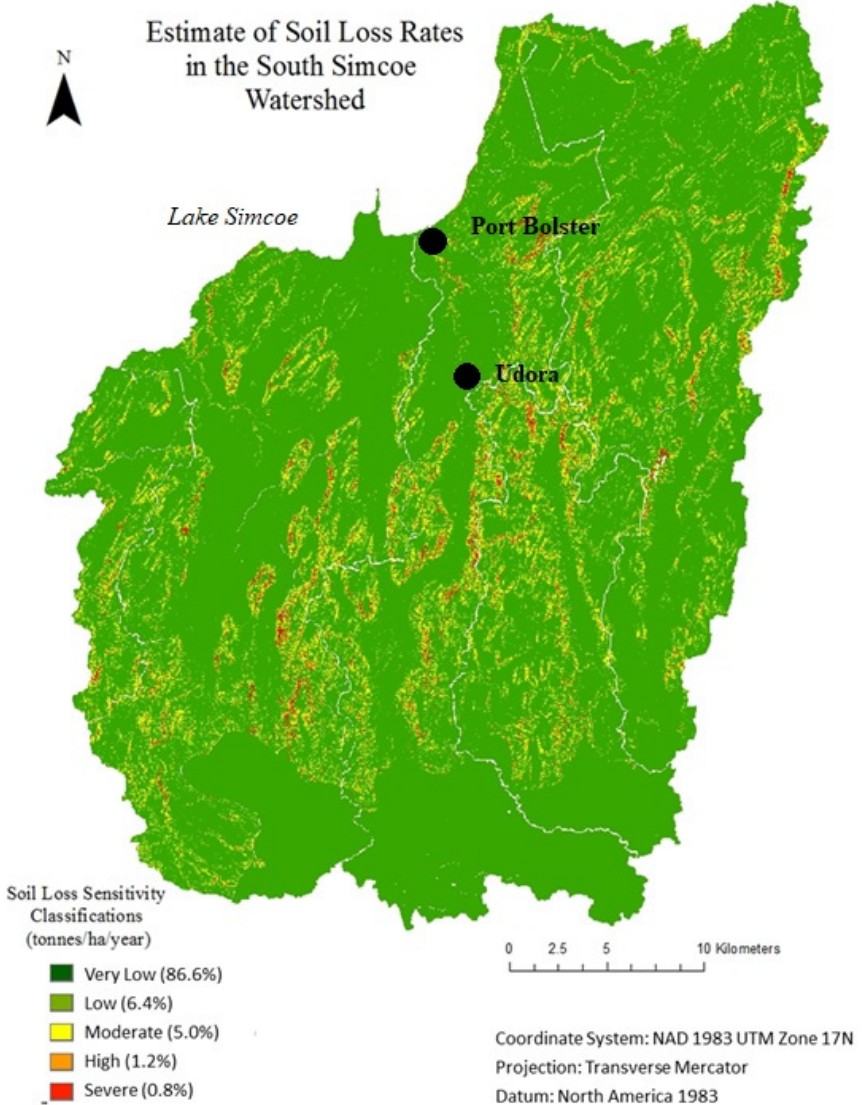

**Figure 5.** Areal extent of the average potential soil loss (t/ha/year) with the Universal Soil Loss Equation.

**Table 1.** Percentage of the area by sub watershed and soil loss severity class.

| Soil Erosion Severity Class | Uxbridge Brook (0) | | Beaver River (1) | | Maskinonge River (2) | | Pefferlaw River (3) | | Black River (4) | | Whole Watershed | |
|---|---|---|---|---|---|---|---|---|---|---|---|---|
| | Total Area | Ag. Area | Total Area | Ag. Area | Total Area | Ag. Area | Total Area | Ag. Area | Total Area | Ag. Area | Total Area | Ag. Area |
| Very Low (0–6.7) tonnes/ha/year | 83.1 | 79.5 | 87.3 | 73.7 | 83.9 | 81.5 | 87.3 | 81.1 | 87.2 | 80.1 | 86.6 | 79.2 |
| Low (6.7–11.2) tonnes/ha/year | 7.9 | 9.9 | 6.5 | 14.1 | 8.1 | 9.3 | 5.6 | 8.6 | 5.9 | 9.2 | 6.4 | 10.2 |
| Moderate (11.2–22.4) tonnes/ha/year | 6.5 | 7.9 | 4.6 | 9.5 | 6.2 | 7.2 | 4.8 | 7.5 | 4.8 | 7.3 | 5.0 | 7.9 |
| High (22.4–33.6) tonnes/ha/year | 1.6 | 1.8 | 1.0 | 1.8 | 1.3 | 1.5 | 1.4 | 2.1 | 1.2 | 1.7 | 1.2 | 1.8 |
| Severe (>33.6) tonnes/ha/year | 0.8 | 0.8 | 0.6 | 0.8 | 0.5 | 0.5 | 0.9 | 0.8 | 0.9 | 0.8 | 0.8 | 0.8 |

### 3.2. USLE and Agriculture Analysis

Agriculture represents a large portion of land use in the south Simcoe Watershed, occupying approximately 49% of the total land cover. Table 1 reveals the proportion of agriculture within each subwatershed occurring on differing soil loss classifications. The preponderance of agricultural operations serendipitously occurs in areas classified with very low sensitivity to soil erosion (Table 1), and a minute proportion of agriculture occurs in areas that are classified as severe soil loss areas. Approximately 79.2, 10.2, 7.9, 1.8, and 0.8% of all agricultural operations across the study site occur in areas classified with very low, low, moderate, high, and severe soil loss, respectively (Table 1).

### 3.3. USLE and Cover Crop Analysis

The application of cover crops within a crop rotation system is an underutilized soil-conservation method in the south Simcoe Watershed, as the presence of cover crops into rotations of corn and soybeans occur on only 18.2% of all agricultural land. This study reveals that agricultural operations incorporate cover crops at a frequency of one year of cover crops grown in last six years (1:6), two years of cover crops grown in the last six years (2:6), three years of cover crops grown in the last six years (3:6), or four years of cover crops grown in the last six years (4:6). Since this study does not consider double-crop management practices and off-season management practices such as red clover, it is unlikely to encounter cover crops grown at frequencies ≥ 4:6, as cover crops are adopted into cash crop sequences and not for monoculture. The most commonly adopted crops are cover crops, at 1:6 frequencies—representing over 90% of cover crop operations (Figure 6). Agricultural areas that have incorporated two, three, and four years of cover crops represent 7.42, 0.57, and 0.004%, respectively.

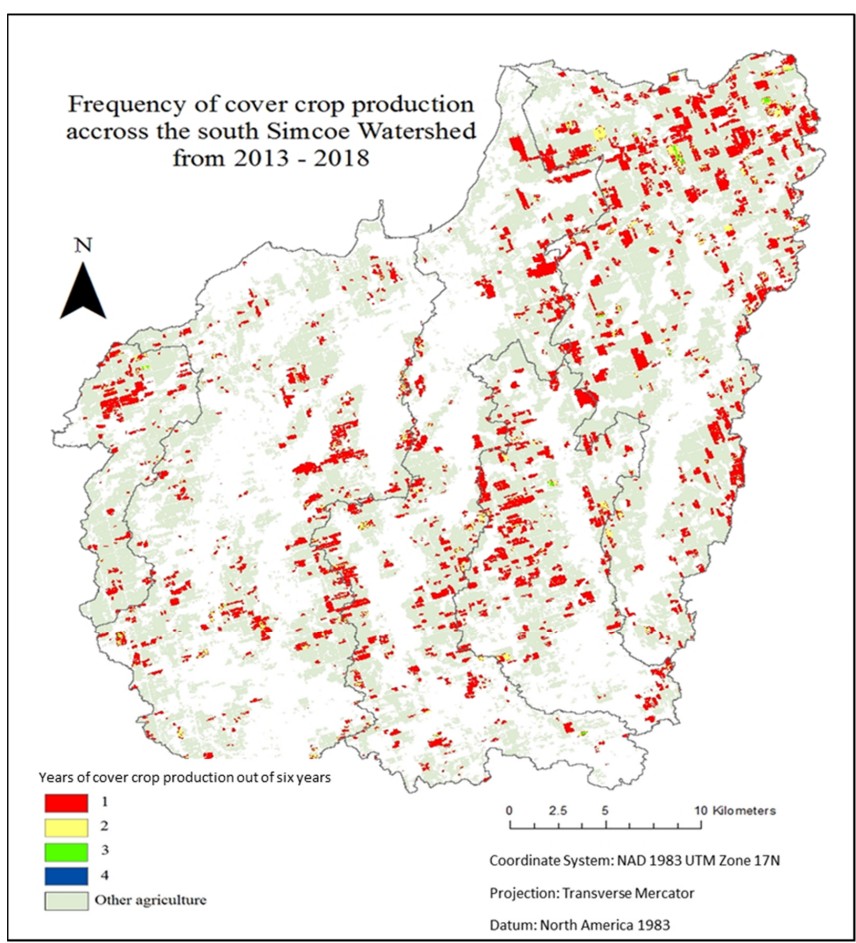

**Figure 6.** Areal extent of cover crop occurrence across the southern Simcoe Watershed from 2013–2018.

The comparative analysis between cover crop adoption and soil loss sensitivity classification reveals that 82% of agricultural productions that have incorporated a cover crop (rye, oats, or winter wheat) at 1:6, 2:6, or 3:6 frequencies over the past six years primarily occur on fields with very low (0–6.7) soil erosion sensitivity. In comparison, 8.7, 6.8, 1.4, and 0.64% represent the portions of agricultural land that has incorporated cover crops over the past six years, on fields classified with low, moderate, high, and severe soil loss, respectively.

Figure 7 shows the relation between the adoption of cover crops (rye, oats, or winter wheat) and soil erosion severity. Our result found that larger areas fall under the high erosion severity class (Figure 7A) compared to other groups that only practice one year of the cover crop out of the six years. In contrast, larger areas fall under the low erosion severity class (Figure 7B and Figure 7C) compared to other groups, in the areas that practiced two to three years of cover crops out of the six years.

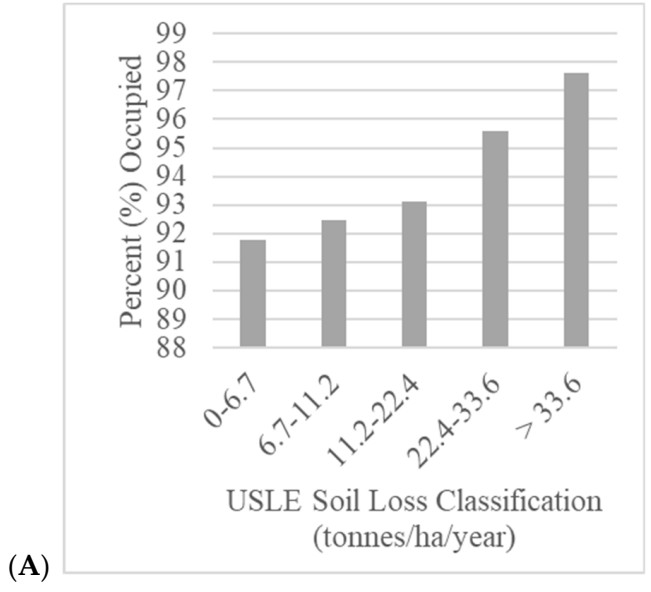

**(A)**

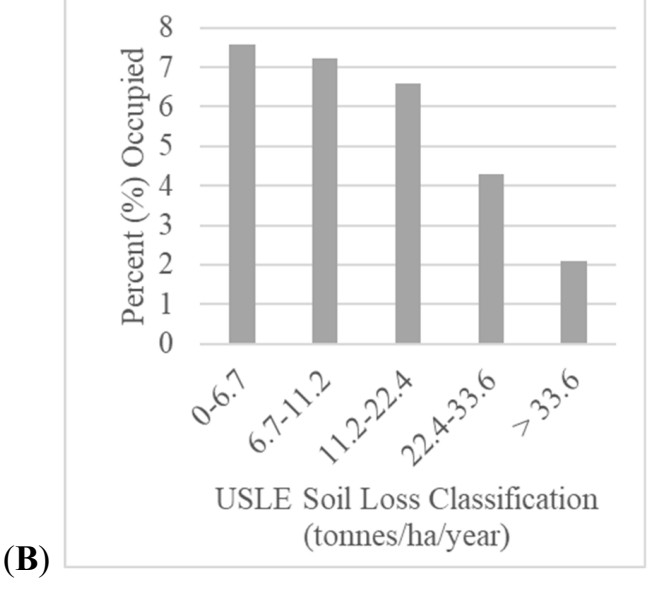

**(B)**

**Figure 7.** *Cont.*

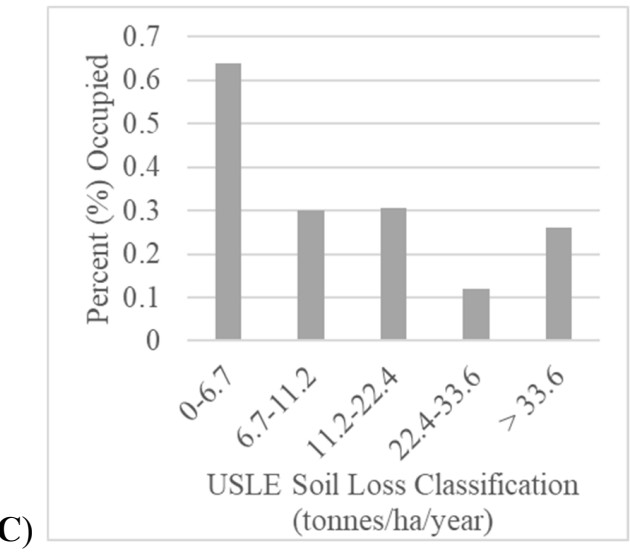

**(C)**

**Figure 7.** Proportion of agriculture that incorporates cover crops at 1:6 (**A**), 2:6 (**B**), 3:6 (**C**) frequencies on soil loss severity.

*3.4. Relationship between Average Soil Loss and Cover Crop Frequency*

In this study, cover crops adopted at higher frequency attribute to lower annual soil loss (Table 2). An average rate of soil loss found for 1:6, 2:6, and 3:6 cover crop frequency is presented in Table 2. The average soil loss for 1:6, 2:6, and 3:6 are approximately 4, 3, and 2 tonnes/ha/year, respectively. This paper suggests that areas incorporating cover crops at a 1:6 frequency produce much higher amounts of soil loss in comparison to 2:6 and 3:6 cover crop adoptions. Currently, agricultural operations incorporating cover crops contribute approximately 40,358 tonnes/ha/year (Table 2); however, if all areas that incorporate cover crops incorporate a 3:6 cover crop frequency, soil loss could be reduced by 17,065 tonnes/ha/year (Table 2).

**Table 2.** Average rate of soil loss for different cover crop frequency classifications.

| Cover Crop Frequency | Total Area (Ha) | Average Rate of Soil Loss (Tonnes/Ha/Year) | Total Soil Loss (Tonnes/Year) | Potential Soil Loss With a 3:6 Cover Crop Frequency (Tonnes/Year) | Diff. (Tonnes/Year) |
|---|---|---|---|---|---|
| 1:6 | 9788 | 4 | 37,880 | 21,436 | 16,444 |
| 2:6 | 786 | 3 | 2343 | 1722 | 621 |
| 3:6 | 61 | 2 | 135 | 135 | 0 |
| **Total** | **10,636** | N/A | 40,358 | 23,293 | 17,065 |

## 4. Discussion

This study reveals that agricultural operations in the Lake Simcoe watershed occur primarily on land with low soil loss sensitivity (Table 2). However, it is not limited to such areas only, as the practice is also extended to areas with severe soil loss sensitivity. Therefore, cover crops are an essential soil-conservation management device that should be adopted, particularly in agriculture fields located on erosion-prone soils. Unfortunately, this study reveals that only 18.2% of areas have incorporated a cover crop over the past six years. Singer et al. [10] reported a similar finding in the US Corn Belt, estimating that 11% of farmers have incorporated a cover crop over five years. Burnett et al. [36] support the lack of cover crop adoption, suggesting that farmers are more inclined to adopt other soil-conservation methods. One possible explanation for the lack of cover crop adoption may be a result of the majority of practices occurring on land that is not prone to soil erosion, thus deterring farmers from adopting cover crops as a soil-conversation practice. There is a need to adopt higher cover crops in areas prone to soil erosion. Limited research is available that

has investigated the direct relationship between cover crop frequency and soil loss rates; however, Gómez et al. [37] found a similar trend to this paper. Gómez et al. [37] compared soil loss rates between conventional tillage with cover crop management in southern Spain. They found that the cover crop management efficiently reduced soil loss compared to conventional tillage. A similar study by Espejo-Péres et al. [38] also investigates soil loss in southern Spain—the results concluded that cover crops diminished soil losses by 76%.

Soil-conservation innovations can be challenging for producers to adopt, as several independent variables persist [39]. Knowler and Bradshaw [39] discuss the benefits and costs of conservation agriculture. Benefits include the reduction of on-farm costs and increase in soil fertility and moisture retention; stabilization of soil protection from erosion; reduction in toxic contamination of surface and groundwater; more regular river flows, reduced flooding and the emergence of dried wells; recharge of aquifers; reduction of air pollution; reduction of atmospheric $CO_2$ emissions; and the conservation of terrestrial and soil-based biodiversity [39]). Interestingly, the discussed benefits are mainly captured by society [39]. In contrast, costs associated with conventional agricultural are concentrated at the farm level. Some of these costs include the purchase of specialized planting equipment, short-term pest problems due to change in crop management, acquiring new management skills, application of additional herbicides, formation and operation of farmers' groups, high perceived risk to farmers due to technological uncertainty, and development of appropriate technical packages and training programs [39]. Similar to Knowler & Bradshaw [39], Burnett et al. [36] discuss socioeconomic and psychological factors influencing farmers' willingness regarding cover crop adoption in the Corn Belt. Burnett et al. [36] suggest that farmers were more inclined to adopt cover crops if they were younger, had a stronger conservation identity, owned more acreage, had less gross farm income, and had a higher response to efficacy. Since conservation agriculture benefits adhere mainly to society, ensuring the benefits at the farm scale over time will promote its adoption [36,39].

Although this research supports previous literature on low adoption rates of cover crops, this research reveals that soil loss rates can be significantly reduced by incorporating cover crops at a higher frequency. Since this study suggests that areas that are prone to soil erosion tend to adopt 1:6 cover crop adoption, these areas need to be targeted to promote higher cover crop frequencies as a soil-conservation practice. By promoting and encouraging the farm-scale benefits of cover crops to operations located in areas prone to soil erosion, a more significant response to adopting cover crops as a soil management practice may be encouraged.

## 5. Conclusions

Cover crops are widely accepted as an efficient soil conservation mechanism to control soil erosion and TP runoff from agriculture during the nongrowing season in Ontario [2,3]. Despite the magnitude of benefits cover crops provide, the adoption response is concerning to operations that remain vulnerable to high erosion. While much recent research investigated cover crop benefits and adoption trends, there remains a gap in the literature to determine the likelihood of cover crop adoption based on the soil erosion sensitivity of the field. This paper has intended to investigate the correlation between cover crop adoption and soil loss severity to understand if soil loss sensitivity plays a significant role in cover crop adoption.

Using AAFC ACI data sets from 2013–2018, cover crop production trends across the south Simcoe Watershed were investigated. The findings, consistent with previous literature, particularly within the Corn Belt of the United States [10,36], suggest that a small portion of agricultural operations adopt cover crop practices. Nevertheless, a more detailed investigation of cover crop frequencies between soil loss sensitivity classifications indicated that there is a consistent adoption trend of incorporating cover crops only 1:6 of the time, despite soil loss sensitivity differences. This study also solidifies the need to target erosion-prone areas by incorporating higher cover crop frequencies, as it significantly reduces soil loss rates in the Simcoe Watershed. Knowler and Bradshaw [39] and

Burnett et al. [36] suggest many farm-level socioeconomic and psychological factors that are deterring producers from incorporating cover crops that may be translated to the south Simcoe Watershed.

In light of the pessimistic results for cover crop adoption in the south Simcoe Watershed, this research reveals the dependence of crop rotation frequency on soil loss sensitivity. Although this dependence has a consistent dominance of low cover crop frequency adoption, by promoting and encouraging cover crop benefits at the farm scale to producers on vulnerable soil, these areas may have a more significant response to more frequently adopting cover crops.

**Author Contributions:** Conceptualization, K.S., K.B.K. and A.B.; methodology, K.S., K.B.K. and A.B.; software, K.S., K.B.K. and A.B.; validation, K.S.; formal analysis, K.S.; investigation, K.S.; resources, K.B.K. and A.B.; data curation, K.S.; writing—original draft preparation, K.S., K.B.K. and A.B.; writing—review and editing, K.S., K.B.K. and A.B.; visualization, K.S.; supervision, K.B.K. and A.B.; project administration, K.B.K. and A.B.; funding acquisition, K.B.K. and A.B. All authors have read and agreed to the published version of the manuscript.

**Funding:** This research was funded by the Food from Thought Program at the University of Guelph, funded by the Canada First Research Excellence Fund (000054) And the APC was funded by Social Sciences and Humanities Research Council (SSHRC) of Canada (430-2019-00094).

**Institutional Review Board Statement:** Not applicable.

**Informed Consent Statement:** Not applicable.

**Data Availability Statement:** All relevant data are within the paper.

**Conflicts of Interest:** The authors declare no conflict of interest.

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
