# Peer review of "Exploring the Relationship between Cover Crop Adoption and Soil Erosion Severity: A Case Study from the Simcoe Watershed, Ontario, Canada"

_land, doi:10.3390/land11070988_

Round 1

Reviewer 1 Report

This paper describes a study that involves using the USLE to calculate soil loss potential in the South Simcoe watershed and linking the results to the spatial pattern of cover crop adaptation. I have attached an annotated version of the manuscript with suggestions for changes and edits, some of which are minor whereas other need a bit more work. Some of the key issues to address are:

Some of the graphs are not professional quality (Fig. 1 and 8, for example)

Results of the statistical analyses are not shown

Tables 1 and 2 could be combined

Figure 6 can be deleted

The paragraph consisting of lines 329 to 345 needs rewriting to provide more clarity

The same goes for the paragraph consisting of lines 371 to 394

The paragraph consisting of lines 395 to 416 would work really well in the introduction.

Author Response

This paper describes a study that involves using the USLE to calculate soil loss potential in the South Simcoe watershed and linking the results to the spatial pattern of cover crop adaptation. I have attached an annotated version of the manuscript with suggestions for changes and edits, some of which are minor whereas other need a bit more work. Some of the key issues to address are:

  • We want to thank you for the endorsement of our paper. We greatly appreciate your valuable time in reviewing our paper. We have revised the paper following your recommendation, including the suggestions and remarks in the annotated version of the manuscript, and believe the revised manuscript is considerably improved, and we hope that it is now deemed ready to publish.  The changes we have made to our manuscript to address your remarks and/or suggestions have been summarized in the paragraphs below.

Some of the graphs are not professional quality (Fig. 1 and 8, for example)

  • Thanks for your remark. Both Figures 1 and 8 are straightforward graphics. Figure 1 is a combination of bar and lines graphic to demonstrate the historical data on discharge with total phosphorous levels in four sub-basins of the Simcoe watershed. And figure 8 illustrates the situation of soil erosion severity in 1, 2, and 3 years of cover crop practiced out the six years. Since both graphics convey a very straightforward message, for this reason, we do believe these simple graphics are fine here. However, considering your suggestions, in the revised version, we made some slight modifications, also considering the other reviewers' remarks on this. Your advice regarding legend, adding the name of the stream in the graph, and stuff related to superscript and abbreviation of average was done as per your suggestions.

Results of the statistical analyses are not shown

  • Yes, you are right. We have removed this part of the text.

Tables 1 and 2 could be combined

  • We followed your suggestion. In the revised manuscript, Table 1 and Table 2 of the initially submitted manuscript have been combined into one. Now it becomes together a Table 1.

Figure 6 can be deleted

  • We agree with your remark that this figure repeats the information already provided in the table. Therefore It has been removed.

The paragraph consisting of lines 329 to 345 needs rewriting to provide more clarity

  • We have Rewritten this paragraph. The main message of this paragraph is to illustrate the situation of soil erosion severity in 1, 2, and 3 years of cover crop practiced out the six years. This part has been written in the revised version of the manuscript as follows: "Figure 8 shows the relation between the adoption of cover crops (rye, oats, or winter wheat) and soil erosion severity. Our result found that larger areas fall under the high erosion severity class (Figure 8a) compared to other groups that only practice one year of the cover crop out of the six years. In contrast, larger areas fall under the low erosion severity class (Figure 8b and Figure 8c) compared to other groups, in the areas that practiced two to three years of cover crops out of the six years."

The same goes for the paragraph consisting of lines 371 to 394

  • We have Rewritten this paragraph. It now reads: "This study reveals that agricultural operations in the Lake Simcoe watershed occur primarily on land with low soil loss sensitivity (Table 2). However, it is not limited to such areas only as the practice is also extended to areas with severe soil loss sensitivity. Therefore cover crops are an essential soil conservation management that should be adopted, particularly on agriculture fields located on erosion-prone soils. Unfortunately, this study reveals that only 18.2% of areas have incorporated a cover crop over the past six years. Singer et al. (2007) reported a similar finding in the US Corn Belt, estimating that 11% of farmers have incorporated a cover crop over five years. Burnett et al. (2015) support the lack of cover crop adoption, suggesting that farmers are more inclined to adopt other soil conservation methods. One possible explanation for the lack of cover crop adoption may be a result of the majority of practices occurring on land that is not prone to soil erosion, thus deterring farmers from adopting cover crops as a soil conversation practice. However, there is an urgent need to adopt higher cover crops in areas prone to soil erosion. Limited research is available that investigated the direct relationship between cover crop frequency and soil loss rates; however, Gómez et al. (2009) found a similar trend to this paper. Gómez et al. (2009) compared soil loss rates between conventional tillage with cover crop management in southern Spain. They found that the cover crop management efficiently reduced soil loss compared to conventional tillage. A similar study by Espejo-Péres et al. (2013) also investigates soil loss in southern Spain – results concluded that cover crops diminished soil losses by 76%.

The paragraph consisting of lines 395 to 416 would work really well in the introduction.

  • We agree with your remark. However, not to lose the logical follow of the reading, we have decided to keep it as it is.

How do you go from the transport index to the slope gradient factor. How do you get the slope length?

  • We have added the explanation as follows: The index combines upslope contributing area (As), under the assumption that contributing area is directly related to discharge, and slope (B).

Does it make sense to use the average value? Peak erosion rates could be associated with bare soil conditions and the average value of the NDVI would not tell you anything about those peak erosion rates.

  • For this study, we believe it makes sense to use average values as it focuses on studying the incorporation of diversifying crops on a year-by-year basis and its relation to soil erosion. Perhaps to address your point, another examination should investigate crop production on a month-by-month basis to address peak erosion rates and the effect of specific crop rotations on a seasonal basis.

Is there any information about what percentage of the study area this would affect?

  • Yes, we have added the following statements in the revised manuscript. This study investigated 49% of the Simcoe Watershed, at 8.7, 6.8, 1.4, and 0.64% of the Agricultural area representing 1:6, 2:6, 3:6, and 4:6 crop rotations, respectively.

How is the expected frequency determined?

  • Expected frequency was determined by multiplying the soil loss/tonne/year for 3:6 by the area of the 1:6, 2:6, and 3:6 plots. However, this section has been removed in the revised manuscript.

Legend is unclear. From the text it is known that the number of of years is out of six, but from the figure you cannot tell that. The figure and caption should include that information.  

  • Fixed

Does not seem to be the right way to refer to this author. Is this K. C. Krishna Bahadur? Use last name(s)

  • In this reference and a few others with a surname “KC”, we would like to inform you that “KC” here is not an initial; it is the legal surname of the author.

  • Thank you very much for your suggestion concerning typos, space, and others. We have corrected them.

Reviewer 2 Report

The article is suitable for publication. But it is necessary to make changes in the description of the methods and improve the figures. You also need to bring the text formatting in accordance with the requirements of the journal. I propose the following changes :

1) Figure 1. This drawing is not clear enough. There are no geographical names on the map. It's awkward to read. You need to increase the dpi of the picture. It is necessary to add the largest rivers and the largest settlements to the map. Also on the map you need a scale bar and a legend.

2) Figure 5 and 7. You need to increase the dpi of the picture. It is necessary to add the largest rivers and the largest settlements to the map.

3) Figure 6. Remove hatching from the drawing. It does not carry any additional information load. Leave only the color. Choose a logical color scale (for example, as in Figure 5).

4) Line 244. Typo. The projection name must contain the number 1984 and not 11984.

5) Paragraph 2.2.4. Include the exact shooting date of the used Lansat shots, as well as patch/row scenes. It can be given in the form of a table. In what software and in what way was the pre-processing of images carried out? Interested in radiometric calibration and atmospheric correction.

6) Correct the formatting of the bibliography.

7) Correct the format of references to literary sources in the text.

Author Response

The article is suitable for publication. But it is necessary to make changes in the description of the methods and improve the figures. You also need to bring the text formatting in accordance with the requirements of the journal. I propose the following changes:

  • First of all, we would like to thank you for the endorsement of our paper. We greatly appreciate your valuable time in reviewing our paper. We have revised the paper following your recommendation and believe the revised manuscript is considerably improved, and we hope that it is now deemed ready to publish. The changes we have made to our manuscript to address your remarks and/or suggestions have been summarized in the paragraphs below.

Figure 1. This drawing is not clear enough. There are no geographical names on the map. It's awkward to read. You need to increase the dpi of the picture. It is necessary to add the largest rivers and the largest settlements to the map. Also on the map you need a scale bar and a legend.

  • Although you mention Figure 1, your remarks seem to be for figure 2 of the initially submitted manuscript. We have produced a revised figure incorporating your suggestion.

2) Figure 5 and 7. You need to increase the dpi of the picture. It is necessary to add the largest rivers and the largest settlements to the map.

  • Done

3) Figure 6. Remove hatching from the drawing. It does not carry any additional information load. Leave only the color. Choose a logical color scale (for example, as in Figure 5).

  • In the revised manuscript, we removed this figure, also considering other reviewers' remarks, as this is just the duplication/repetition of the information already presented in a Table.

4) Line 244. Typo. The projection name must contain the number 1984 and not 11984.

  • Done

5) Paragraph 2.2.4. Include the exact shooting date of the used Lansat shots, as well as patch/row scenes. It can be given in the form of a table. In what software and in what way was the pre-processing of images carried out? Interested in radiometric calibration and atmospheric correction.

  • It's a great suggestion. However, because all the paper's authors are either on vacation or on fieldwork, we cannot access the metadata of the Landsat image used for this study. Therefore unable to spell out the information you are asking us to elaborate on. We are incredibly sorry for this.

6) Correct the formatting of the bibliography.

  • Done

7) Correct the format of references to literary sources in the text.

  • Done

Reviewer 3 Report

       In the current manuscript, the author studied the relationship between cover crop adoption frequency and the soil erosion severity in a watershed. Generally, when the researchers proposed a new technique or a management practices, they want the farmers or the policymaker to use it. This manuscript reported a very interesting result, most of the farmers did not adopt this good practices (cover crop), or they did not use it in the right area. This result proposed an important issue about how the soil erosion can be controlled in an area when we already owned a good technique. In this case, the socio-economic driving factor for the formation and maintenance of ecosystem service must be considered. Sometimes, the lack of adoption of new technique is due to the cost; sometimes, it is caused by the fact that they do not know how and where to use it. It seems that both of those two reasons occurred in the current study, and I guess it is also applicable for other places. The English is good. Below are some specific comments.

1.       Ln44: you have told that the soil erosion and phosphorus loadings occurred mainly at the non-growing season in the introduction section, I think this judgement is right and reasonable, thus, I don’t think it is necessary to present those results here.

2.       Ln 107-122: you do not need so many examples.

3.       Ln 270: can you tell me how the expected frequency was calculated? Based on the assumption that the areas with higher soil erosion severity should adopt a higher cover crop frequency?

4.       Ln 274: are you sure the one-way ANOVA analysis was correct here? I guess this kind of statistical analysis was used to detect the treatment effect among different groups.

5.       Table 2: the title is not very clear. Do you mean the percentage of farmland distributed in different classifications? Since “soil loss tolerance” was only present here, maybe you can delete this concept here.

6.       Ln 348-351: those three sentences showed very similar meaning. Condense it.

7.       Ln 379-383: can you explain it more clearly? Do you mean that the farmers thought it is unnecessary to plant cover crops in the area that is not prone to soil erosion? I think it is reasonable. In addition, no more frequency of cover crop adoption occurred in the area with a higher soil erosion severity, which is contradicted to our hypothesis. It is interesting. Can you tell me something about this management practices in your place? Why the farmers adopt this practices? Due to the policy reason? Or they though this practices is helpful for controlling soil erosion, and they want to do it. Or the farmers don’t know the overall pattern of soil erosion in your studied area, thus certainly they don’t know which area is more urgent to adopt this management practices?

8.       Ln 397-409: good explanation. The farmer paid for this management practices, but it benefit all the people who did not pay for it directly. You can reorganized those sentences more concisely.

Overall, this manuscript need a revision before acceptance.

Author Response

In the current manuscript, the author studied the relationship between cover crop adoption frequency and the soil erosion severity in a watershed. Generally, when the researchers proposed a new technique or a management practices, they want the farmers or the policymaker to use it. This manuscript reported a very interesting result, most of the farmers did not adopt this good practices (cover crop), or they did not use it in the right area. This result proposed an important issue about how the soil erosion can be controlled in an area when we already owned a good technique. In this case, the socio-economic driving factor for the formation and maintenance of ecosystem service must be considered. Sometimes, the lack of adoption of new technique is due to the cost; sometimes, it is caused by the fact that they do not know how and where to use it. It seems that both of those two reasons occurred in the current study, and I guess it is also applicable for other places. The English is good. Below are some specific comments.

  • First of all, we would like to thank you for the endorsement of our paper. We greatly appreciate your valuable time in reviewing our paper. We have revised the paper following your recommendation and believe the revised manuscript is considerably improved, and we hope that it is now deemed ready to publish. The changes we have made to our manuscript to address your remarks and/or suggestions have been summarized in the paragraphs below.

  1. Ln44: you have told that the soil erosion and phosphorus loadings occurred mainly at the non-growing season in the introduction section, I think this judgement is right and reasonable, thus, I don't think it is necessary to present those results here.
    • Ok

  1. Ln 107-122: you do not need so many examples.
    • Ok

  1. Ln 270: can you tell me how the expected frequency was calculated? Based on the assumption that the areas with higher soil erosion severity should adopt a higher cover crop frequency?

  • As we mentioned above, to respond to a remark from Reviewer 1 that expected frequency was determined by multiplying the soil loss/tonne/year for 3:6 by the area of the 1:6, 2:6, and 3:6 plots. However, this section has been removed in the revised manuscript.

  1. Ln 274: are you sure the one-way ANOVA analysis was correct here? I guess this kind of statistical analysis was used to detect the treatment effect among different groups.

  • Statistical analysis was, in fact, not needed to demonstrate the relationship between cover crop adoption and soil erosion severity, which was shown through a series of maps, graphics, and descriptive information. For that reason, we have remoted all the statistical test-related text discussed in the initially submitted manuscript version.

  1. Table 2: the title is not very clear. Do you mean the percentage of farmland distributed in different classifications? Since "soil loss tolerance" was only present here, maybe you can delete this concept here.
    • Thank you for your great suggestion. In the revised manuscript, we combined the original table 1 and table 2 and also produced a revised title.

  1. Ln 348-351: those three sentences showed very similar meaning. Condense it.
    • In the revised manuscript, this part of the text has been removed.

  1. Ln 379-383: can you explain it more clearly? Do you mean that the farmers thought it is unnecessary to plant cover crops in the area that is not prone to soil erosion? I think it is reasonable. In addition, no more frequency of cover crop adoption occurred in the area with a higher soil erosion severity, which is contradicted to our hypothesis. It is interesting. Can you tell me something about this management practices in your place? Why the farmers adopt this practices? Due to the policy reason? Or they though this practices is helpful for controlling soil erosion, and they want to do it. Or the farmers don't know the overall pattern of soil erosion in your studied area, thus certainly they don't know which area is more urgent to adopt this management practices?

  • In the revised manuscript, this part has been rewritten. It now reads: "This study reveals that agricultural operations in the Lake Simcoe watershed occur primarily on land with low soil loss sensitivity (Table 2). However, it is not limited to such areas only as the practice is also extended to areas with severe soil loss sensitivity. Therefore cover crops are an essential soil conservation management that should be adopted, particularly on agriculture fields located on erosion-prone soils. Unfortunately, this study reveals that only 18.2% of areas have incorporated a cover crop over the past six years. Singer et al. (2007) reported a similar finding in the US Corn Belt, estimating that 11% of farmers have incorporated a cover crop over five years. Burnett et al. (2015) support the lack of cover crop adoption, suggesting that farmers are more inclined to adopt other soil conservation methods. One possible explanation for the lack of cover crop adoption may be a result of the majority of practices occurring on land that is not prone to soil erosion, thus deterring farmers from adopting cover crops as a soil conversation practice. However, there is an urgent need to adopt higher cover crops in areas prone to soil erosion. Limited research is available that investigated the direct relationship between cover crop frequency and soil loss rates; however, Gómez et al. (2009) found a similar trend to this paper. Gómez et al. (2009) compared soil loss rates between conventional tillage with cover crop management in southern Spain. They found that the cover crop management efficiently reduced soil loss compared to conventional tillage. A similar study by Espejo-Péres et al. (2013) also investigates soil loss in southern Spain – results concluded that cover crops diminished soil losses by 76%.

  1. Ln 397-409: good explanation. The farmer paid for this management practices, but it benefit all the people who did not pay for it directly. You can reorganized those sentences more concisely.

  • In the revised manuscript, we have rewritten some parts of this, also incorporating another reviewer's remark.

Overall, this manuscript need a revision before acceptance.

  • We revised the manuscript incorporating all the suggestions, including the other two reviewers.